# Morphology-modified contributions of electronic transitions to the optical response of plasmonic nanoporous gold metamaterial

Tlek Tapani[1,11], Jonas M. Pettersson[1,11], Nils Henriksson[1,11], Carla M. Brunner[1,11], Ann Céline Zimmermann[1], Erik Zäll[2], Nils V. Hauff[3], Lakshmi Das[1], Anastasiia Sapunova[4,5], Gianluca Balestra[6,7], Massimo Cuscunà[6], Aitor De Andrés[1], Tommaso Giovannini[8], Denis Garoli[4,9] & Nicolò Maccaferri[1,10] ✉

Nanoporous metals have emerged as promising functional architectures with tunable optical and electronic properties, high surface areas, and applicability in sensing, catalysis, and biomedicine. While their linear optical behavior and morphological properties have been extensively studied, the electronic properties, and in particular how they are affected by morphology, remain not fully understood. Here we combine experimental and theoretical studies of electronic excitation and relaxation in a nanoporous gold metamaterial. Optical pump–probe experiments show slower electron relaxation dynamics compared to the continuous film, consistent with a higher transient electronic temperature and stronger smearing of the Fermi–Dirac distribution, well reproduced by an extended two-temperature model. Furthermore, cathodoluminescence measurements reveal broadband localized plasmon resonances, and atomistic simulations disentangle intra- and interband effects, demonstrating that nanoscale porosity fundamentally reshapes the electronic response. These findings support nanoporosity as a key design parameter for controlling steady-state and ultrafast optical behavior in plasmonic materials.

Nanoporous metals are well established materials with capability to enhance light-matter interactions[1]. They enable applications in many areas, including nanophotonics[2,3], biomedicine[4–8], spintronics[9], and energy conversion[10]. Notably, they can be considered metamaterials, as their nanostructured surface volume can be designed on purpose to have properties which are not found in ordinary materials[11]. In fact, it has been shown that nanoporous structures behave as plasmonic metamaterials whose effective plasma frequency can be tuned by controlling the fractal dimensions, i.e., ligament and pore morphology[1,12–14]. Furthermore, the presence of nanoscale pore-like structures and grain boundaries introduces additional degrees of freedom for tuning plasmon-induced phenomena[15,16], making nanoporous metals attractive platforms for developing efficient hot-electron generators[17] and optical filters[18,19], where controlled optical absorption is crucial. In this context, nanoporous gold (NPG) films have emerged as promising systems due to their tunable optical

[1]Ultrafast Nanoscience Group, Department of Physics, Umeå University, Umeå, Sweden. [2]Nano for Energy Unit, Department of Physics, Umeå University, Umeå, Sweden. [3]Umeå Centre for Electron Microscopy, Umeå University, Umeå, Sweden. [4]Istituto Italiano di Tecnologia, Genova, Italy. [5]Department of Materials Science, University of Milano-Bicocca, Milan, Italy. [6]CNR NANOTEC Institute of Nanotechnology, Lecce, Italy. [7]Department of Mathematics and Physics 'Ennio de Giorgi', University of Salento, Lecce, Italy. [8]Department of Physics, University of Rome Tor Vergata, Rome, Italy. [9]Dipartimento di Scienze e Metodi dell'ingegneria, Università di Modena e Reggio Emilia, Reggio Emilia, Italy. [10]Wallenberg Initiative Materials Science for Sustainability, Department of Physics, Umeå University, Umeå, Sweden. [11]These authors contributed equally: Tlek Tapani, Jonas M. Pettersson, Nils Henriksson, Carla M. Brunner. ✉e-mail: nicolo.maccaferri@umu.se

properties[20], high surface area and interconnected nanostructures, making them highly versatile for applications in sensing[21–23] and catalysis[24,25]. Nevertheless, the processes underlying their functionalities are not yet fully understood, and the possibility of capturing the inner mechanisms might allow for engineering their electronic and optical properties with increased precision.

It is well known that gold (Au) exhibits interband transitions from the 5d band to the conduction 6sp band[22] in the visible spectral range close to 2.4 eV (~520 nm). In the context of plasmonics, previous research has thus been primarily focused on how interband and intraband transitions influence plasmonic behavior[26–28], and only recently, studies have pointed out the role of interband and intraband transitions on plasmonic excitations at ultrafast timescales[29–33]. Interestingly, while interband transitions are typically regarded as loss and passive channels in plasmonic systems[34], their potential active role in ultrafast carrier generation remains an open question[35–37]. Similarly, while it is known that nanoporous materials effectively act as better catalysts due to the larger surface to volume ratio, a more comprehensive understanding of their electronic properties can support progress in optimization of ultrafast and/or plasmon-supported catalytic processes.

Here, we perform an integrated experimental and theoretical study unveiling the role that morphology plays on intraband and interband transition contributions to the optical response of an NPG metamaterial. First, we investigate the ultrafast relaxation dynamics of thermalized hot carriers in bulk gold (BG) and NPG via nondegenerate pump-probe spectroscopy. We observe that, in the NPG film, the transient transmission ($\Delta T/T$) signal associated with interband transitions from the 5d band to the 6sp conduction band occurs at photon energies below 2.3 eV, significantly lower than the onset energy observed in BG[38]. By modeling the transient transmission spectroscopy measurements with an analytical approach based on the extended two-temperature model (e2TM), we attribute this effect to an enhanced generation of hot carriers in NPG providing more empty states in the 6sp conduction band, and consequently requiring lower photon energy for interband electron excitation from the 5d band compared to BG. Furthermore, cathodoluminescence (CL) spectroscopy measurements indicate the presence of localized plasmon excitations in NPG from 2.4 eV (L-point of Au band structure) to 1.8 eV (X-point of Au band structure). These experimental results are underpinned by atomistic electrodynamics simulations based on the fully atomistic frequency-dependent fluctuating charges and dipoles (ωFQFμ) model, which allows us to disentangle intraband and interband transitions contributions to the optical response of both BG and NPG cases.

Our results provide physical insights on how morphology alters the electronic dynamics in NPG. Such understanding is important for developing engineered plasmonic platforms based on NPG where control over carrier generation and relaxation is essential, including in hot carrier driven processes, as for instance in light harvesting and catalysis[39,40].

## Results

NPG films were synthetized on fused silica substrates using a recently developed dry method[2,41] (for more details see "Methods"). This approach enables the preparation of NPG thin films completely free of impurities from other metals, a typical limitation observed in the standard preparation techniques based on dealloying[1]. As benchmark sample, a BG film thermally evaporated on fused silica was also fabricated. X-ray photoelectron spectroscopy (XPS) measurements were performed on all samples, confirming that no oxidation occurred, in particular in NPG (see Supplementary Note 1 and Supplementary Fig. 1). Representative images of the NPG are reported in Supplementary Note 1 and Supplementary Fig. 2.

### Transient response and numerical modeling

First, we performed pump-probe measurements (for more details, see "Methods", Supplementary Note 2 and Supplementary Figs. 3 and 4) to understand the ultrafast carriers' dynamics upon photoexcitation by a 12-fs infrared pump pulse (central wavelength 850 nm). The dynamics were probed by a broadband pulse of (duration 11 fs), covering a spectral range from 500 to 750 nm. The ultrafast optical responses of both NPG and BG films are presented in Fig. 1a, b, d, e. $\Delta T/T$ measurements showing the evolution of the signal over the time delay between pump and probe pulses and across different wavelengths are displayed in Figs. 1a, d for BG and NPG films, respectively.

Upon pump excitation of Au, hot electrons undergo fast thermalization via electron-electron scattering on timescales on the order of 10 fs[42]. Electron-phonon scattering leads to heat exchange of the electron bath with the lattice, exhibiting typical relaxation times on the ps timescale[34,43]. Electronic density redistribution around the Fermi level manifests in a change of the dielectric permittivity and thus of the transmission, which is experimentally measured via the broadband probe pulse. The NPG film shows a more intense $\Delta T/T$ signal characterized by a broad negative dip spanning from 540 to 670 nm (Fig. 1d). This feature is absent in the BG film at the same fluence of 3 mJ cm$^{-2}$, where a negative $\Delta T/T$ signal is present only at wavelengths shorter than 570 nm, consistent with the presence of interband transitions at the L-point of the Au band structure, confirming previous studies on BG films[38,44].

Figure 1b shows $\Delta T/T$ traces of BG (purple curve) and NPG (yellow curve) films at 540 nm, (purple and yellow dashed lines in Fig. 1a, d) up to 10 ps (longer time traces up to 20 ps are shown in Supplementary Fig. 5). A larger variation of the $\Delta T/T$ signal together with longer relaxation dynamics can be clearly observed in the NPG film case, consistent with reports in previous studies[17,45,46]. This observation is a result of a combination of factors. Firstly, the morphology of NPG leads to a decrease of the difference of the effective refractive index of the sample and surrounding air and therefore to an increased absorption at the pump wavelength[47] (see Supplementary Note 3 and Supplementary Fig. 6). Consequently, the initial electron temperature rises higher in NPG (see inset in Fig. 1b). The maximum variation of the electronic gas temperature in the BG (purple curve) and NPG (yellow curve) was estimated by the e2TM (see "Methods" for more details) to be ~800 K and ~3200 K, respectively. Secondly, the heat capacity of Au increases with electronic temperature[48], and higher heat capacity slows down relaxation dynamics. Thus, the elevated initial electron temperature in NPG compared to BG is responsible for the slower dynamics. A single exponential fit on the pump-probe traces in Fig. 1b quantifies this contrast as an increase of relaxation time of 3.1 ps for NPG compared to only 1 ps for BG, which seems reasonable if we consider that the electronic temperature in NPG is roughly 3 times that reached in BG for the same pump fluence.

We now move our focus to the spectral dependence of the $\Delta T/T$ signals of BG (green curve) and NPG (pink curve) films (Fig. 1e), which correspond to the green and pink horizontal dashed lines in the maps in Fig. 1a, d, at a time delay of 430 fs. At this time delay the $\Delta T/T$ signal is maximum in both NPG and BG cases. The BG film exhibits a negative signal around 520 nm, indicating increased absorption due to interband transitions. This result is consistent with previous reports on ultrafast dynamics of bulk Au[38,44]. In comparison, the NPG film displays a broader negative signal, suggesting that transient interband transitions can take place in an extended spectral region following pump excitation. This broadening can be understood as a result of the following mechanism: under steady-state conditions at room temperature, the occupied electronic states follow a Fermi−Dirac distribution around the Fermi level $E_F$. Both intraband transitions involving electronic states around $E_F$ and interband transitions between the occupied 5d and empty states in the 6sp band can take place. After pump

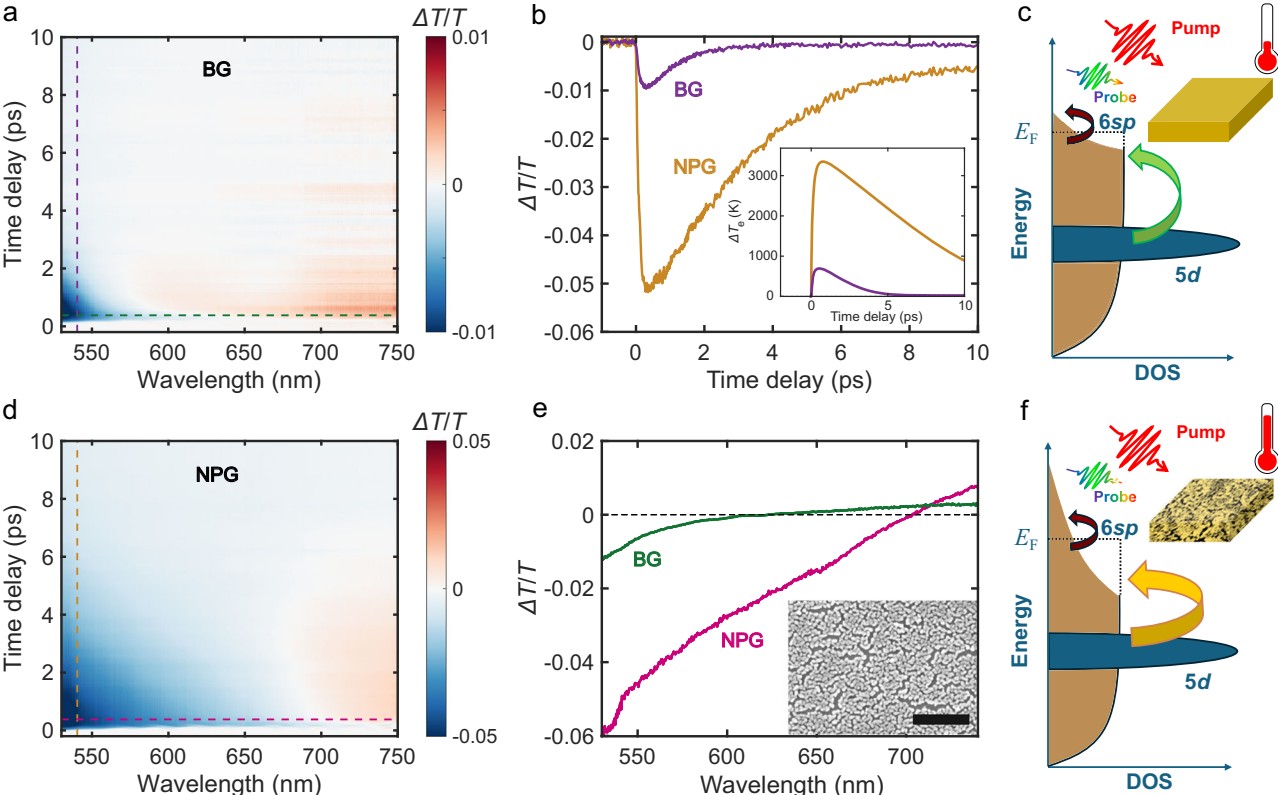

**Fig. 1 | Experimental transient optical response of bulk and nanoporous Au.**
**a** Transient transmission ($\Delta T/T$) of bulk gold (BG) as a function of the time delay between pump and probe pulses (vertical axis) and wavelength of the probe pulse (horizontal axis). **b** $\Delta T/T$ of BG (purple curve) and nanoporous gold (NPG, yellow curve) films as a function of the time delay between pump and probe pulses at 540 nm. Inset: transient variation of the electronic temperature in BG (purple curve) and NPG (yellow curve) calculated using the extended two-temperature model (e2TM). **c** Sketch of density of states (DOS) vs. energy, showing the main mechanism behind the ultrafast response of BG, involving contributions from

intraband transitions (dark red arrow) and interband transitions (green arrow).
**d** $\Delta T/T$ of NPG. **e** $\Delta T/T$ of BG (green curve) and NPG (pink curve) films as a function of the probe light wavelength at a time delay of 430 fs. Inset: SEM image of the NPG sample (scale bar: 200 nm). **f** Sketch showing the main mechanism behind the ultrafast response of NPG, involving contributions from intraband transitions (dark red arrow) and interband transitions (yellow arrow). Compared to BG, the elevated electron temperature leads to stronger redistribution of occupied states around the Fermi level, thus allowing for interband transitions at lower energies.

excitation and fast electronic thermalization, a stronger redistribution around the Fermi level frees electronic states below $E_F$, an effect that is considerably more pronounced in NPG than in BG due to higher electronic temperature, as sketched in Fig. 1c, f.

In BG, interband transitions typically require energies exceeding 2.3 eV (green arrow in Fig. 1c). In contrast, in NPG, the significantly elevated electron temperature results in a greater availability of empty states in the $6sp$ band below $E_F$, thus enabling increased absorption of photons at energies below 2.3 eV (yellow arrow in Fig. 1f). Therefore, the negative transient transmission signal measured in NPG extends over longer photon wavelength (lower energy).

For a more detailed characterization of the mechanism sketched in Fig. 1c, f, we employed the e2TM considering the geometry of the NPG via the Bruggeman effective medium approximation (EMA). Additionally, we modeled the permittivity variation as a function of the electron and lattice temperatures, in combination with the transfer matrix method (TMM). This model allows for the calculation of transient transmission signals as presented in Fig. 2 (for more details see "Methods" and Supplementary Note 4). Our calculations reproduce the experimental observations accurately, corroborating the conclusion that the mechanism outlined above, which is implemented in our model, is responsible for the transient transmission change. Thus, we can attribute the observed transient effects in NPG to the morphology of the nanoporous structure that is quantitatively well represented by the filling factor, i.e., the volumetric solid fraction of Au in the nanoporous film. It is worth mentioning here that, while the EMA

description does not account for the presence of localized plasmonic excitations in NPG, these excitations may contribute to the enhanced absorption[47] observed experimentally (see Supplementary Note 3 and Supplementary Fig. 6), thus providing the system with more energy upon pump pulse excitation. Nevertheless, such an increased plasmonic absorption does not significantly affect the physics of the mechanism investigated with our model, which is mainly connected to the porosity of the system[43,49] and well described by the EMA.

For our simulations of BG and NPG films, we assume that both systems are excited by a pulse centered at 850 nm with a fluence of 3 mJ cm⁻², matching the experiments reported in Fig. 1a, d. We consider a film thickness of 30 nm in both cases, and a filling factor of 0.4 was used in the NPG case according to the SEM images (see Supplementary Fig. 2). In Figs. 2a, c, we display the calculated $\Delta T/T$ signal as a function of the pump-probe time delay (vertical axis) and the wavelength of the probe pulse (horizontal axis) for the BG and NPG films, along with the transient transmission data at the vertical and horizontal cuts in Fig. 2b, d, analogous to Fig. 1. The transient transmission of the NPG film displays a longer relaxation time compared to the BG film case, confirming that our model captures the underlying physical mechanisms responsible for the spectral features observed in the experiments. Moreover, the calculated negative $\Delta T/T$ response in the NPG film is broader, also reproducing the main experimental features.

In further support of the identified mechanism, where the deposited energy primarily determines relaxation dynamics, we performed additional experiments with varying pump fluence. At higher

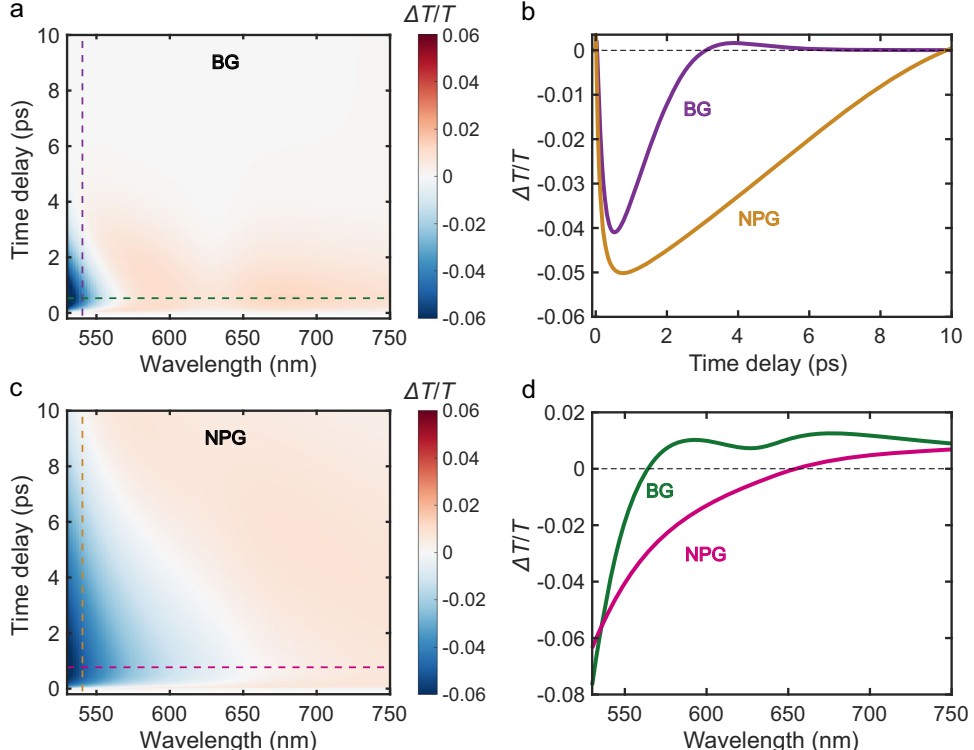

**Fig. 2 | Simulations of the ultrafast optical response of bulk and nanoporous Au.** Calculated transient transmission ($\Delta T/T$) of bulk gold (BG) film (**a**) and nanoporous gold (NPG) film (**c**), as a function of time delay between pump and probe pulses (vertical axis) and wavelength of the probe light pulse (horizontal axis) using the extended two-temperature model (e2TM), Bruggeman effective medium approximation (EMA) and the transfer matrix method TMM. **b** $\Delta T/T$ of BG (purple) and NPG (yellow) films as a function of pump probe time delay at 540 nm. **d** $\Delta T/T$ of BG (green) and NPG (pink) films as a function of probe wavelength at time delays corresponding to the signal minima (530 fs for BG, and 770 fs for NPG).

incident pump power, the electronic temperature directly following excitation increases, leading to increased smearing of the Fermi-Dirac distribution, which in turn raises the probability that probe light with lower photon energy will promote interband transitions below 2.3 eV. In NPG, this effect is expected to be stronger with respect to BG, since its smaller geometric heat capacity drives a larger increase in electron temperature for the same energy deposited onto the system. Consequently, we expect the short wavelength negative $\Delta T/T$ feature to deepen and broaden with higher fluence, especially in NPG. Results from the fluence-dependent study on both films are presented in Fig. 3.

In Fig. 3a, b, we show the measured $\Delta T/T$ signal of BG and NPG as a function of the probe light wavelength for different pump pulse fluences spanning from 1 to 4 mJ cm$^{-2}$, with step of 0.5 mJ cm$^{-2}$, at fixed pump-probe time delay of 430 fs. Indeed, the magnitude of the negative signal increases with fluence and shifts to longer wavelengths, in particular in the case of the NPG film (see black arrow in Fig. 3b). The experimental results match the predictions obtained from our model, shown in Fig. 3c, d. Notably, the model reproduces both the increase in amplitude and the pronounced broadening of the negative $\Delta T/T$ signal with increasing fluence, in particular for the NPG film case, thus supporting our claim that the transient optical response of the NPG film arises mainly from thermal effects, which are increased by the morphology of the NPG film.

It is also worth mentioning here that our simulations confirm that the negative $\Delta T/T$ signal is indeed due to an increase of the imaginary part of the dielectric permittivity, which is the quantity associated with absorption. This rules out a more trivial origin of the observed effects, such as positive transient reflection larger than $\Delta T/T$, which would not imply that electronic transitions are promoted by the probe light (see

Supplementary Note 5 and Supplementary Fig. 7). The transient change of the real and imaginary part of the permittivity upon pump excitation is shown in Supplementary Fig. 7. The imaginary part is positive for both BG and NPG, indicating that the observed effects arise from probe light absorption and not from ground state bleaching (which would give a negative transient absorption signal). Notably, the absorption in NPG is much stronger also below 1.8 eV (X-point in the Au electronic band structure) also for long time delays (>1 ps).

## Linear response

Furthermore, to assess whether a change in morphology affects the band structure of the NPG system, we measured XPS valence-band spectra of both BG and NPG (see Supplementary Note 6 and Supplementary Fig. 8). Within the resolution of our instrument (0.1 eV) we do not detect any change in the electronic structure, thus supporting our conclusion that the observed differences between NPG and BG arise from macroscopic effects, i.e., porous morphology, rather than changes in the electronic band structure itself.

Having shown that nanoporosity modifies the ultrafast optical response of the sample, and that this is connected to a modified interband transition response under femtosecond pump excitation, we investigated whether this is a purely transient effect or whether it also appears in steady-state conditions. We performed CL spectroscopy (see details in "Methods" section), and the results are reported in Fig. 4. In the BG case, the CL map (Fig. 4b), which shows the wavelength of maximum CL intensity, is nearly uniform. Notably, the CL spectra (Fig. 4c) corresponding to the colored crosses in Fig. 4a, show a clear peak near 540 nm, the onset of the 5d to 6sp interband transition in bulk Au (close to the L-point in Au band structure). On the other hand, NPG exhibits a markedly different response: the CL map for NPG in

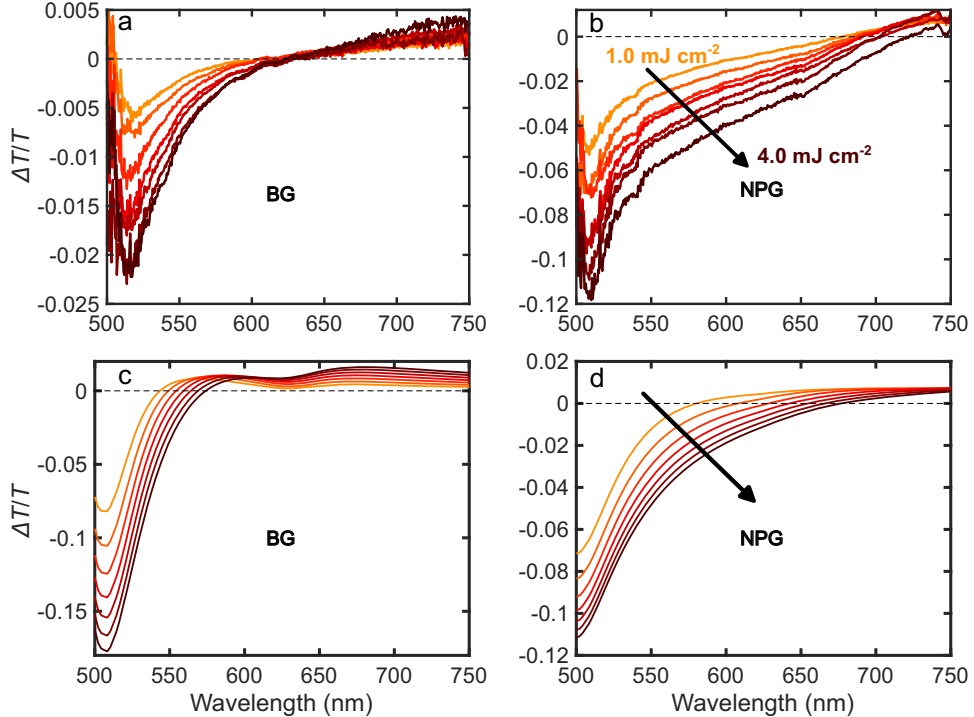

**Fig. 3 | Fluence dependence of transient transmission in bulk and nanoporous Au.** Experimental (**a**, **b**) and theoretical (**c**, **d**) pump fluence dependence of transient transmission ($\Delta T/T$) of bulk gold (BG) film (**a**, **c**) and nanoporous gold (NPG) film (**b**, **d**), with fluence varying from 1 to 4 mJ cm$^{-2}$ in steps of 0.5 mJ cm$^{-2}$.

Fig. 4e shows a very broad variation in signal caused by the high anisotropy and dispersion in size of the metallic clusters in NPG films. Figure 4f confirms the spatially dependent emission behavior, displaying the CL spectra at the positions marked by the colored crosses in Fig. 4d, revealing a broad set of localized excitations due to the presence of localized plasmon resonances centered around 600 nm, except from the gap regions with emission in the blue (see blue curve in Fig. 4f, corresponding to the CL signal coming from a region where Au is either almost or entirely absent). Additionally, from Fig. 4d, which is a SEM image of the NPG sample, it can be observed that the interconnecting wire size is comparable to the mean free path of electrons in Au, that is around 50 nm[50]. In the context of ultrafast dynamics, this may lead to increased electron-surface scattering. However, our model suggests that this effect is negligible compared to the impact of the higher electron temperature in NPG following excitation (for more details see "Methods"). It is worth mentioning here that in our linear optical response measurements (see Supplementary Fig. 6), we collect the signal from a much larger area, thus we probe a response averaged over many of these localized modes, resulting in a nearly flat absorption spectrum. Interestingly, in this spectral range, localized plasmon resonances in Au are known to have contributions from both intraband and interband transitions[51,52].

**Atomistic modeling**
To disentangle these two contributions and understand with more detail the effect of morphology on the optical behavior measured in our pump-probe and CL measurements, we used the fully atomistic frequency-dependent fluctuating charges and dipoles ($\omega$FQF$\mu$)[53–56] method. This approach has been shown to accurately reproduce the optical response of metal nanostructures from time dependent density functional theory calculations but at much lower computational cost, which is essential when considering large (more than hundreds atoms) systems[57]. The model is based on the assignment of time-varying charges and dipole moments to each atom, associated with intraband and interband transitions, respectively. In the frequency

domain, the method is reformulated in terms of complex charges and dipole moments, oscillating at a single frequency (for additional details see "Methods"). We have summarized the simulation results in Fig. 5. We have studied two model systems ($8 \times 8 \times 2$ nm$^3$) to emulate the response of a BG film and an NPG structure (see Fig. 5a). The NPG model structure (3590 atoms) was obtained by random atom removal of the BG model structure (6844 atoms). The wavelength-dependent polarizability $\alpha_z$ is derived from the imaginary part of the total dipole moment induced by an incident electric field $E_z$ and shown in Fig. 5b. The simulated optical behavior agrees well with the CL measurements above (see Fig. 4), as compared to the BG (blue) film, showing a peak around 520 nm, the NPG model system (orange) exhibits several peaks between 550 nm and 720 nm.

Additionally, employing this simulation method, we have introduced a metric to quantify the contributions from intraband ($\langle f_q \rangle$, purple) and interband ($\langle f_\mu \rangle$, red) effects to the optical response (see Fig. 5c) based on the calculated charge and dipole densities ($\rho_q, \rho_\mu$) (see "Methods" for more details). We can observe that for both BG and NPG, the interband contributions $\langle f_\mu \rangle$ dominate at shorter wavelengths up to 600 nm ($\langle f_\mu \rangle \gg \langle f_q \rangle$). At longer wavelengths, the NPG system retains a mixed character ($\langle f_\mu \rangle \approx 0.54$, $\langle f_q \rangle \approx 0.46$ at 700 nm) that clearly differentiates it from the BG case where $\langle f_q \rangle \gg \langle f_\mu \rangle$. This result highlights a stronger interband role in the optical response of NPG. To visually understand this contrast between the BG and NPG film cases, we display the interband and intraband contributions to the optical response at 700 nm in Fig. 5d. For the BG film case (left panel in Fig. 5c), the charge density $\rho_q$ due to intraband contributions is almost identical to the total induced charge and dipole moment density, while the interband contribution is oscillating out of phase ($\langle f_\mu \rangle < 0$), as expected in d-metals nanostructured materials[58]. These results show how morphology, even when varied at the atomistic level, is affecting the electronic behavior of the samples under investigations, thus pointing out the crucial role of nanoporosity in shaping the relative contributions of intraband and interband transitions to both the transient and steady state optical behavior in NPG metamaterials.

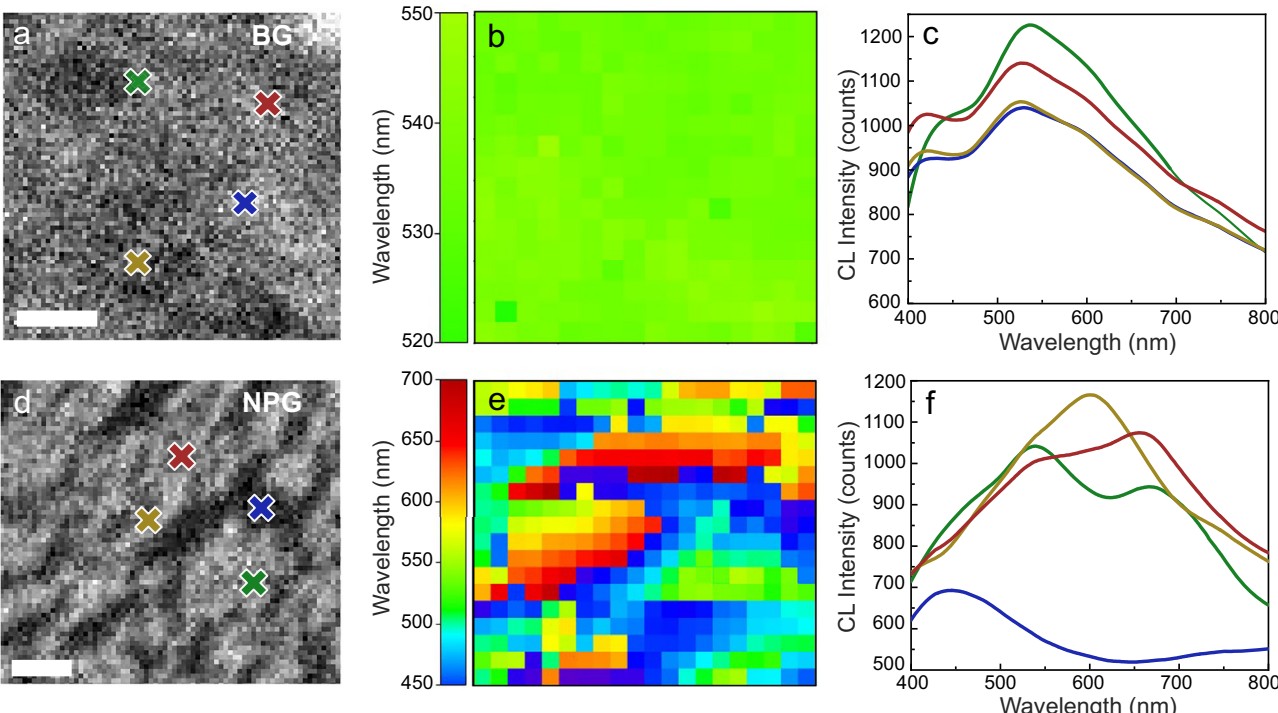

**Fig. 4 | Cathodoluminescence spectroscopy of bulk and nanoporous Au.**
**a** Scanning electron microscopy (SEM) image of bulk gold (BG); scale bar: 100 nm.
**b** Cathodoluminescence (CL) wavelength-dependent intensity map of the region in
(**a**), showing an almost uniform response with a peak at 540 nm. **c** CL spectra at the
colored crosses in (**a**), featuring a broad peak near the conventional interband
threshold (~540 nm). **d** SEM of nanoporous gold (NPG); scale bar: 100 nm. **e** CL
wavelength-dependent intensity map for the region in (**d**), revealing strong spatial
variations characteristic of localized plasmon modes. **f** CL spectra at the colored
crosses in (**d**), displaying position dependent resonances spanning the visible range
(500–700 nm), consistent with localized plasmon resonances excitation due to the
sample morphology.

## Discussion

In this work we reported that the interaction of light with Au via
interband transitions can be modified by sample morphology,
namely nanoscale porosity. Firstly, we present the experimental
observation of increased transient interband transition probability
in an NPG metamaterial thin film by using ultrafast pump-probe
spectroscopy. While our benchmark BG film exhibits the char-
acteristic negative transient transmission signal at the well-known
interband transition around 540 nm (2.3 eV), the NPG film displays
a significantly broader response, suggesting that, due to porosity,
interband transitions at lower energies contribute to the optical
response, resulting in increased probe light absorption at these
wavelengths. A computational approach based on the e2TM com-
bined with the Bruggeman EMA and the TMM, which considers the
geometry of the porous material, fully reproduces the experi-
mental transient transmission spectroscopy measurements, sup-
porting the conclusion that porosity is indeed crucial in modifying
the contributions of electronic transitions to the optical response
of NPG compared to the BG case. This effect can be explained by
increased absorption per metal volume in NPG, resulting in a higher
electronic temperature in the system upon pump excitation and
thus leaving more empty states at lower energies in the 6sp band,
allowing interband transitions from the 5d band with probe pulse
photons of energy below 2.3 eV. Fluence-dependent measurements
confirm this description, as increasing pump power raises the
electronic temperature and broadens the negative spectral region
in transient transmission due to lower energy interband transitions.
Interestingly, the applied opto-thermal workflow is controlled by
morphology and can be generalized to colloidal or assembled
films, suggesting the phenomenon may occur across a broader
class of systems (see Supplementary Note 7 and Supplemen-
tary Fig. 9).

Additionally, CL measurements show that NPG displays localized
excitations spanning the visible spectral range between the L- and
X-point of the Au band structure, a behavior that is absent in BG. These
excitations, related to the presence of localized plasmon resonances,
contain contributions from both intra- and interband transitions. To
disentangle these contributions, we employed atomistic ωFQFμ
simulations, showing that the NPG model system supports multiple
resonances in the 550–700 nm range consistent with CL observations.
These simulations also demonstrate that, unlike the BG system, NPG
retains comparable intraband and interband contributions, associated
with charge and dipole density distributions, to the optical response at
longer wavelengths below the X-point of the Au band structure.

Overall, our results have a direct implication in hot carrier gen-
eration, as the increased availability of low-energy empty states in the
conduction band offers more interband excitation pathways in a
broader range of energies spanning the visible and near-infrared
spectral region. Furthermore, generation of hot carriers from either
intraband and interband transitions can be engineered to tailor ther-
mionic injection of charges in either molecules or semiconductors
before phonon thermalization – an "active" channel for photocatalysis
and photodetection[59]. In the context of NPG metamaterials, we can
control these properties by systematically varying the filling factor. In
fact, this parameter may be used to tune the interband cut-off wave-
length, as shown in Supplementary Note 8, where we calculated the
$\Delta T/T$ signal as a function of the probe wavelength and filling factor
(see Supplementary Fig. 10). We are now confident that morphology
plays an important role in ultrafast electronic dynamics and tuning
intra- and interband transitions, offering an additional degree of con-
trol over light-matter interactions at ultrafast timescales. These
insights open opportunities for applications of nanoporous materials
in many fields, such as plasmon-driven catalysis, ultrafast photo-
chemistry, and active nanophotonics.

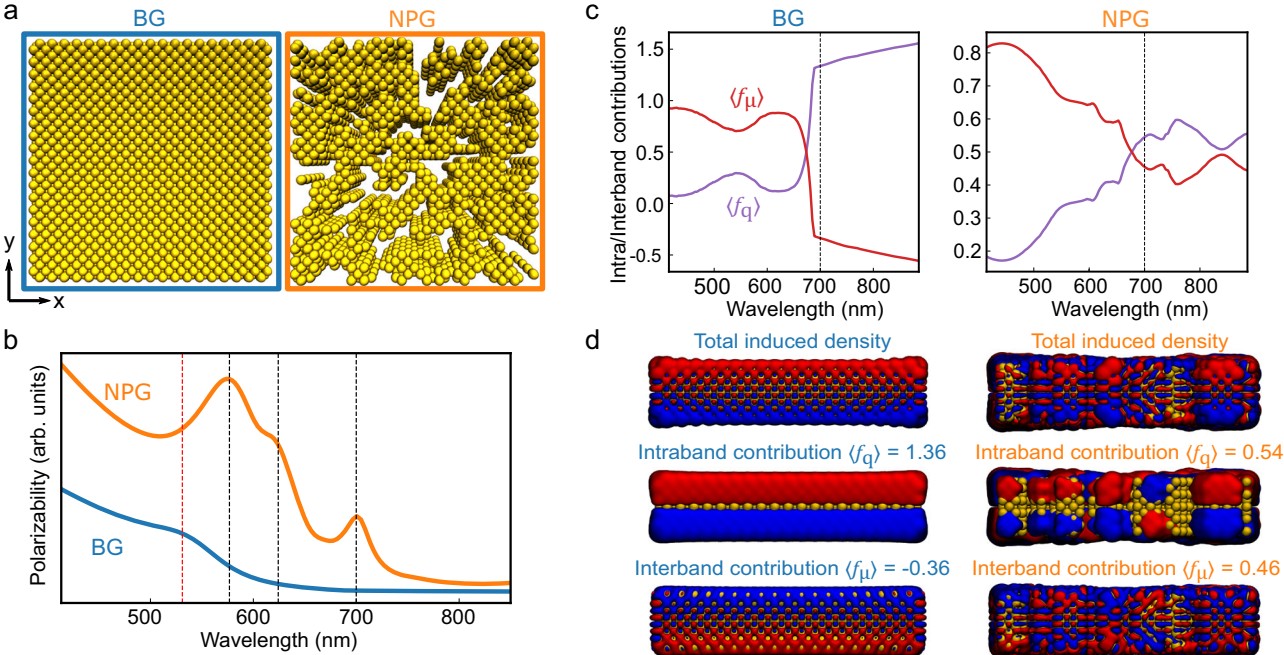

**Fig. 5 | Atomistic electrodynamic modeling of bulk and nanoporous Au.**
**a** Model structure for bulk gold (BG, left) and nanoporous gold (NPG) system (right) exploited in ωFQFμ simulation. **b** Calculated imaginary polarizability spectra (zz-component). The z-axis as polarization direction of the incident light field was chosen since the response along this axis does not suffer from finite-size effects observed along the x- and y- directions in the spectral range of interest. NPG exhibits multiple resonances as indicated by the black dashed lines, whereas BG shows a broader response with a single resonance at 525 nm as indicated by the red dashed line. **c** Wavelength-dependent intraband and interband contributions ($\langle f_q \rangle$ and $\langle f_\mu \rangle$, respectively) to the induced charge and dipole densities for the BG (left panel) and the NPG (right panel) systems, obtained from the ωFQFμ decomposition. Above 700 nm (black dashed lines) NPG retains a mixed character, while intraband contributions dominate for the BG model system. **d** Intraband and interband contributions to the total induced density in BG (left panel) and NPG (right panel) systems at 700 nm.

## Methods

### Sample fabrication

NPG thin films were prepared following a dry process method introduced previously in refs. [2,41]. In summary, a Poly(methyl methacrylate) (PMMA) thin layer was spin-coated on a fused silica substrate and baked at 180 °C for 3 min. Then, a thin Au layer was deposited by electron-beam evaporation using a tilted configuration (i.e., placing the sample tilted at 80° with respect to the evaporation source). The process was completed with the etching of the PMMA layer by means of $O_2$ plasma (200 W, 10 min). A BG thin film was prepared using standard electron-beam evaporation on a fused silica substrate. Both the NPG and BG films have a nominal thickness of 26 nm. Due to its morphology change during fabrication[41], the NPG film has a varying thickness around this value (±4 nm, that is a maximum of 8 nm from dip to peak).

### Pump-probe measurements

The pump-probe employs a Yb:KGW laser (model: PH2-20W-SP, Light Conversion) to generate both pump and probe pulses. The probe pulse was generated through a 6-mm-thick YAG crystal (undoped YAG, orientation [100]), and compressed using custom-made dielectric chirped mirrors (Ultrafast Innovation). These pulses have a spectrum in the range 500–750 nm (2.5–1.3 eV) with 11 fs duration and repetition rate of 50 kHz. The pump was generated using a commercial Visible to Near-Infrared Optical Parametric Amplifier (VIS-NIR OPA, model: ORPHEUS-N-2H, Light Conversion), producing a spectrum in the range 730–885 nm (1.7–1.4 eV) with 12 fs duration. For more details on the characterization of pump and probe pulses see Supplementary Note 2. An external pulse picker was adopted to halve the repetition rate of the pump beam to 25 kHz for differential transmission ($\Delta T/T$) measurements. A spectrograph (model: HSVIS camera with sensor HA:S14290Q-1024, Stresing) was used to collect the transmitted probe

intensity that provides two-dimensional $\Delta T/T$ maps as a function of pump-probe delay and probe wavelength. A tunable intensity filter (model: NDC-25C-4-B, Thorlabs) was also added for fluence-dependence measurements.

### Cathodoluminescence measurements

NPG films were fabricated on a 100 nm thick $Si_3N_4$ membrane to minimize emission from bulky substrates during measurements, which were performed at room temperature by using a Zeiss Merlin scanning electron microscope equipped with a CL imaging system (model: SPARC, Delmic). CL spectra were collected in the range 400–800 nm with an "Andor Kimera 193i" spectrometer with a focal length of 193 mm and a grating of 300 gr mm$^{-1}$. The photon emission was captured by an "Andor Newton DU920P-BEX2DD" CCD camera with a maximum quantum efficiency of 90%. The electron beam operated at an acceleration voltage of 30 kV and an emission current of 10 nA. The color-coded map consisted of 20 × 18 pixels (pixel size: 25 nm). The focused electron beam was scanned across the specimen, dwelling for 10 s (integration time) on each pixel to acquire the CL spectra.

### Numerical modeling of transient response

The full simulation of a pump-probe experiment was done in three steps[60,61]: first by (i) calculating the steady-state pump-matter interaction, followed by (ii) calculating the transient change in permittivity induced by the pump, and lastly (iii) the probe interaction with the excited structure. This requires steady-state calculations of reflection, absorption and transmission, and subsequent modeling of the electron and lattice temperature change induced by the pump at different time delays. Then, the transient permittivity change is calculated as a function of the time delay, after which the steady-state calculations are repeated with the altered permittivity at all time delays. Here below we report all these steps in more detail.

(i) Abeles TMM[62] was used to calculate the steady state reflection ($R$), transmission ($T$) and thus absorption ($A = 1 - T - R$) of NPG and BG on a glass substrate (see dashed lines in Supplementary Fig. 6). The geometry is a semi-infinite layer of air, followed by either the BG or the NPG film with a finite thickness (in this case the thickness was assumed to be 30 nm in both cases), and finally a semi-infinite layer of glass. We considered the incident light to imping on the structure at normal incidence from the air side. Au permittivity from a Drude-Lorentz model[63] was used in the calculations. For NPG, we utilized the Bruggeman EMA to treat the NPG as a film with an effective permittivity[49,64–66]. This model is valid for samples with relatively high filling factors (>30%) where there is no clear host material, as is the case for NPG. The effective permittivity is determined by solving the equation[64]

$$\sum_{i=1}^{n} f_i \frac{\varepsilon_i - \varepsilon_{\mathrm{Br},p}}{\varepsilon_{\mathrm{Br},p} - \nu_{i,p}(\varepsilon_i - \varepsilon_{\mathrm{Br},p})} = 0 \tag{1}$$

where $n$ is the number of different materials in the NPG (in this case $n = 2$, as the system is made of air and gold). The index $i = 1$ represents Au, while $i = 2$ represents air. $\varepsilon_{\mathrm{Br},p}$ is the effective permittivity in directions $p$, $\varepsilon_{\mathrm{Au}}$ the bulk Au permittivity and $\nu_{i,p}$ the depolarization coefficient for the $i$-th inclusion in direction $p$. The depolarization coefficients depend on the geometry of the inclusions. We assume that the porosity is the same in all direction, thus we use $\nu_{i,p} = 1/3$ for both Au and air, and in all directions $p$. Therefore, the effective permittivity is isotropic. Equation (1) was solved numerically for each wavelength of interest, and the physical solutions with $\mathrm{Im}(\varepsilon_{\mathrm{Br}}) > 0$ were selected. In Supplementary Fig. 5, the calculated and measured steady state transmission, reflection and absorption of both samples are shown.

(ii) To model the temperature change induced by the pump, we utilized the extended two-temperature model (e2TM), which was first implemented by Carpene[67]. However, an instantaneous excitation and a homogeneous spatial distribution of the electron and lattice heat temperatures was assumed for simplicity, as we focused on the slower relaxation dynamics[17] governed by electron-phonon coupling. This version of the e2TM is represented by the following system of equations

$$\begin{aligned} C_{\mathrm{e}}(T_{\mathrm{e}})\frac{\partial T_{\mathrm{e}}}{\partial t} &= -G_{\mathrm{ep}}(T_{\mathrm{e}})(T_{\mathrm{e}} - T_{\mathrm{l}}) + \frac{\partial U_{\mathrm{ee}}}{\partial t} \\ C_{\mathrm{l}}\frac{\partial T_{\mathrm{l}}}{\partial t} &= G_{\mathrm{ep}}(T_{\mathrm{e}})(T_{\mathrm{e}} - T_{\mathrm{l}}) + \frac{\partial U_{\mathrm{ep}}}{\partial t} \end{aligned} \tag{2}$$

where $t$ is the time delay between pump and probe, $T_{\mathrm{e,l}}$ are the electron and lattice temperatures, $C_{\mathrm{e}}(T_{\mathrm{e}})$ the temperature dependent electronic volumetric heat capacity, $C_{\mathrm{l}}$ the lattice volumetric heat capacity and $G_{\mathrm{ep}}(T_{\mathrm{e}})$ the electron-phonon coupling factor. We refer the readers to the original work by Carpene[67] for a detailed derivation of the source terms $\partial_t U_{\mathrm{ee}}$ and $\partial_t U_{\mathrm{ep}}$. Under the instantaneous excitation assumption, which in this case is reasonable being the dynamics studied longer than 500 fs and the pump and probe pulses are 15 fs long, the source terms are calculated as

$$\frac{\partial U_{\mathrm{ee}}}{\partial t} = -\exp\left(-\left(\tau_{\mathrm{ee}}^{-1} + \tau_{\mathrm{ep}}^{-1}\right)t\right)\left[t + \tau_{\mathrm{ee}}\left(1 - \exp(-\tau_{\mathrm{ee}}^{-1}t)\right)\right]t^{-2}P \tag{3}$$

$$\frac{\partial U_{\mathrm{ep}}}{\partial t} = -\exp\left(-\left(\tau_{\mathrm{ee}}^{-1} + \tau_{\mathrm{ep}}^{-1}\right)t\right)\left[\tau_{\mathrm{ee}}\left(1 - \exp(-\tau_{\mathrm{ee}}^{-1}t)\right)\right]t^{-1}\tau_{\mathrm{ep}}^{-1}P \tag{4}$$

where $\tau_{\mathrm{ee,ep}}$ represents the electron-electron and electron-phonon scattering rates. Finally, $P = AF/d$, where $A$ is the absorption of the sample calculated in step (i) at the pump wavelength, $F$ the fluence of the pump, and $d$ the thickness of the sample. Here, the absorbed energy is assumed to be distributed equally across the thickness of the

**Table 1 | Values of parameters used for simulations**

| Parameter | Value | Reference |
|---|---|---|
| $G_{\mathrm{ep}}$ | Fitting | 48 |
| $C_{\mathrm{e}}$ | Fitting | 48 |
| $C_{\mathrm{l}}$ | $2.46 \times 10^6$ J m$^{-3}$ K$^{-1}$ | 76 |
| $\tau_0$ | 3.4 fs | 67 |
| $\tau_{\mathrm{ep}}$ | $1.4 \times 10^3$ fs | 67 |
| $E_{\mathrm{F}}$ | 7.3 eV | 77 |

The table displays the values of the simulation parameters, or if the values were acquired from fitting of data, together with references.

sample. The electron-electron scattering rate was determined as[67]

$$\tau_{\mathrm{ee}} = \tau_0 \left(\frac{E_{\mathrm{F}}}{E_{\mathrm{pu}}}\right)^2 \tag{5}$$

where $\tau_0$ is a constant, $E_{\mathrm{pu}}$ the pump photon energy and $E_{\mathrm{F}}$ the Fermi energy of Au. Table 1 shows the values used for each parameter. Since we treated the NPG as an effective medium that is a mixture of air and gold, where only the gold volume fraction contributes to the heat capacity and electron-phonon coupling, the parameters in Eq. (2) had to be scaled to account for the sample morphology. The heat capacity of air is negligible compared to that of Au and does not meaningfully contribute to the effective heat capacity of NPG. Since the volumetric heat capacities are used in Eq. (2), the effective values for the NPG are the Au values scaled with the filling factor $f$, such that only the Au part of the NPG contributes to the heat capacities. The same reasoning is applied to the electron-phonon coupling factor. Electrons only interact with phonons inside the Au, therefore the electron-phonon coupling factor, which is a volumetric quantity, also scales linearly with $f$. Previous studies of NPG films of smaller ligament size have indicated an increase of the electron-phonon coupling factor compared to the BG counterpart[68,69]. However, since the electron-phonon coupling primarily involves short-wavelength phonons[70], we neglected surface-mediated energy transfer from electrons to the lattice, which we assume is a secondary effect in our NPG sample that has an average ligament size of ~50 nm. The negligible contribution of this effect is confirmed by our results in the main text (compare Fig. 1, experiment, and Fig. 2, theory). So, to represent NPG in the e2TM, we scale the parameters as $C_{\mathrm{e,l}} \rightarrow fC_{\mathrm{e,l}}$ and $G_{\mathrm{ep}} \rightarrow fG_{\mathrm{ep}}$. Mathematically, this is equivalent to scaling the source terms with $f^{-1}$, which means that the energy absorbed from the pump is distributed over less volume of Au in the NPG case compared to the BG film. Using the temperatures from the e2TM, we can estimate the permittivity change $\Delta\varepsilon(T_{\mathrm{e}}, T_{\mathrm{l}})$ induced by the pump. The change in the imaginary part of the Au permittivity induced by the pump was determined through semi-classical modeling of optical transitions in Au using hyperbolic approximations of the band structure[60,61,71]. This change is related to interband transitions and changes due to Fermi smearing of the bands. The corresponding real part was determined using Kramers-Kronig relations. Additionally, the variation of permittivity induced by intraband changes following pump-excitation was also estimated, see Supplementary Note 4 for more details.

(iii) Since the NPG is represented by the Bruggeman EMA, the procedure of simulating the probe interaction was adjusted for NPG compared to BG. For BG, steady-state simulations using the TMM, as described above, were performed for each time delay $t$ with Au permittivity $\varepsilon_{\mathrm{Au}}(t) = \varepsilon_{\mathrm{Au}} + \Delta\varepsilon(T_{\mathrm{e}}(t), T_{\mathrm{l}}(t))$. The permittivity variation $\Delta\varepsilon$ is complex and includes both the intraband and interband contribution described above. For NPG, we assumed that the permittivity of the air remains unchanged after pump excitation, and thus only the Au

permittivity change contributes to a change in the variation of the effective permittivity. $\varepsilon_{Au}(t)$ was therefore computed in the same way as for BG but with temperatures change considering the scaling factor $f$, and the effective permittivity $\varepsilon_{Br}(t)$ was determined using Eq. (1) with the altered gold permittivity.

## Atomistic modeling

We employed the atomistic frequency-dependent fluctuating charges and dipoles (ωFQFμ) model[57]. In this model, both a complex charge $q_i$, associated with intraband effects, and a complex dipole moment $\mu_i$, accounting for interband transitions[54], are assigned to each metal atom $i$. The charge exchange between the atoms is governed by a Drude conduction mechanism and modulated by a phenomenological description of quantum tunneling[53], which is essential for the description of gaps, junctions, and defects, as in the case of NPG[54]. To describe d-metals, the complex dipole moment is introduced, this way taking into account the polarizability of the d-shell[54,58]. The charge and dipole distributions at each atomic site for an external electric field $\mathbf{E}^{ext}$ oscillating at frequency $\omega$ are obtained by solving the following complex coupled linear equations[54]

$$-i\omega q_i(\omega) = \frac{2n_0\tau}{1-i\omega\tau}\sum_j\left[1-f\left(l_{ij}\right)\right]\frac{A_{ij}}{l_{ij}}\left(\phi_j^{el}-\phi_i^{el}\right) \quad (6)$$

$$\boldsymbol{\mu}_i = \alpha_i^\omega\left(\mathbf{E}_i^{ext}+\mathbf{E}_i^\mu+\mathbf{E}_i^q\right) \quad (7)$$

where $n_0$ is the electron density, $\tau$ the friction (damping) time, $A_{ij}$ the effective area connecting atoms $i$ and $j$ and $l_{ij}$ their distance. $\phi^{el}$ denotes the electrochemical potential acting on each metal atom. It accounts for interatomic interactions (charges and dipole moments) and their coupling to the external electric field. The function $f\left(l_{ij}\right)$ is a Fermi-like function that mimics quantum tunneling. $\mathbf{E}^\mu$, and $\mathbf{E}^q$ are the local fields generated by induced dipoles and by charges, respectively. Finally, $\alpha(\omega)$ is the complex, frequency-dependent atomic polarizability used to represent interband contributions, which can be obtained from the experimental permittivity[54]. Solving these equations allows to describe the optical response of noble-metal nanoparticles with ab initio–level accuracy[54], at a much reduced computational cost. The model can also be reformulated in the time domain, enabling the study of the time evolution of the plasmon decay under a time-dependent excitation[72]. The ωFQFμ scaling efficiency can be exploited to run simulations of systems containing thousands of atoms[20]. Here, we build a BG structure ($8\times8\times2$ nm$^3$; lattice parameter 4.08 Å) comprising of 6844 atoms. The NPG structure is then generated by randomly removing columns of Au atoms from the BG lattice, yielding a structure containing 3590 atoms (see also Fig. 5a).

Owing to its formulation based on well-defined physical quantities, ωFQFμ allows for dissecting the intra- and interband contributions, which are associated with charge and dipole terms, respectively, to the induced total charge and dipole density, as

$$\rho(\mathbf{r}) = \rho_q(\mathbf{r}) + \rho_\mu(\mathbf{r}) \quad (8)$$

where

$$\rho_q(r) = \sum_i \frac{q_i}{\pi^{3/2}R_{q_i}^3}\exp\left(-\frac{|\mathbf{r}-\mathbf{r}_i|^2}{R_{q_i}^2}\right) \quad (9)$$

and

$$\rho_\mu(r) = \sum_i \frac{|\boldsymbol{\mu}_i|}{\pi^{3/2}R_{\mu_i}^3}\hat{\mathbf{n}}_i\cdot\nabla_{\bar{r}_i}\exp\left(-\frac{|\mathbf{r}-\mathbf{r}_i|^2}{R_{\mu_i}^2}\right) \quad (10)$$

Here, $\rho_q, \rho_\mu$ represent charge and dipole moment density after summation over all atoms $i$. $R_q$ and $R_\mu$ are the Gaussian widths of the charges and dipoles, respectively, mimicking a diffuse quantum-like density[73]. To quantify intra- and interband contributions, we introduce the following metric, averaging respective contributions from the complete structure:

$$\langle f_X\rangle = \frac{\int_\Omega f_X(\mathbf{r})|\rho(\mathbf{r})|\,d^3\mathbf{r}}{\int_\Omega |\rho(\mathbf{r})|\,d^3\mathbf{r}}\quad X = \{q,\mu\} \quad (11)$$

where $f_X(\mathbf{r})$ is defined as

$$f_X(\mathbf{r}) = \begin{cases} 0, & \rho(\mathbf{r})=0 \\ \frac{\rho_X(\mathbf{r})}{\rho(\mathbf{r})}, & \rho(\mathbf{r})\neq0 \end{cases}\quad X = \{q,\mu\} \quad (12)$$

Note that, by definition $\langle f_q\rangle + \langle f_\mu\rangle = 1$. All ωFQFμ calculations were performed by using the open-source code plasmonX[57], using the Au parameters defined in ref. 54.

## Data availability

The simulation and transient transmission data generated in this study have been deposited in the figshare database under accession code https://doi.org/10.6084/m9.figshare.30737969[74].

## Code availability

The code used for the simulations is publicly available with an MIT license on Zenodo under accession code https://doi.org/10.5281/zenodo.18175029[75].

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

## Acknowledgements

This work was funded by the Swedish Research Council (Grants No. 2021-05784 and No. 2025-04734, N.M.), the Knut and Alice Wallenberg Foundation (Grant No. 2023.0089, N.M.), the European Research Council (ERC Starting Grant No. 101116253 "MagneticTWIST", N.M.), the European Union's Horizon 2020 Research and Innovation Programme under the Marie Skłodowska-Curie Actions (Grant No. 101147248, L.D.), the HORIZON-Pathfinder-Open "3D-BRICKS" (Grant No. 101099125, D.G.) and the HORIZON-MSCADN-2022 "DYNAMO" (Grant No. 101072818, D.G.). The authors thank Dr. Andrey Shchukarev and Dr. Dmitry Shevela from the XPS Platform at the Department of Chemistry, Umeå University, for the support with XPS measurements and analysis. The authors acknowledge Umeå Centre for Electron Microscopy (UCEM) and the National Microscopy Infrastructure (NMI), for instrument access and technical support.

## Author contributions

T.T., J.M.P., A.C.Z. and A.D.A. performed the pump-probe measurements. N.H. developed the theory with input from T.T., C.M.B. and N.M. D.G. performed the sample fabrication with the support of A.S. G.B. and M.C. performed the cathodoluminescence measurements. E.Z., J.M.P., and C.M.B. performed linear absorption measurements. N.V.H. and L.D. performed SEM characterization. T.G. performed atomistic simulations. T.T., J.M.P., N.H., C.M.B., T.G. and N.M. analyzed and discussed the results. T.T., N.H., C.M.B., and N.M. wrote the manuscript with contributions from all the authors. N.M. conceived the study and supervised the work.

## Funding

## Competing interests
The authors declare no competing interests.
