## [Transparent Peer Review file · Nature Communications]

Morphology-modified contributions of electronic transitions to the optical response of plasmonic nanoporous gold metamaterial

Corresponding Author: Professor Nicolò Maccaferri

Version 0:

Reviewer comments:

Reviewer #1

(Remarks to the Author)

Tapani et al. studied the transient response of a nanoporous metal gold (NPG) substrate using pump-probe spectroscopy, and contrast it with that of a flat gold layer (BG). Their experimental study is explored using a computational model combining transfer matrix method (TMM) and a two-temperature model (2TM) to account for the absorbance of the system and internal electronic dynamics, respectively. The analysis of the results focuses on the possibility of allowing interband transitions in gold after an initial pulse elevates the effective electronic temperature and consequently creates empty states in the conduction band at energies lower than the Fermi level.

The article explores an interesting phenomenon, but I think that it does not explore it in sufficient depth: some key aspects of the characterization do not seem to match the discussion and explanation of the results, the theoretical modelling is not sufficiently well described and fails to explain one of the significant features of the data. Although I would be very interested in seeing a significantly extended version of this study to better understand the phenomenon at hand, I do not think that this manuscript is ready for publication.

1) The difference between BG and NPG on their initial effective electron temperature after the pulse is argued to arise from a difference in their absorption at the pump's wavelength, 850 nm. However, not only this difference is not shown in the main text or SI, but the spectral data shown in Fig. S2, which is presented as justification for the larger absorption of the NPG, suggests instead that their absorption might not be that different after all. Cathodoluminescence (CL) data, which could perhaps be used as an indirect proxy for the absorbance, does not reach these wavelengths either.

2) On the 2T model, additional information is needed: (i) explicit details about the values of beta taken from Ref. 19 (for instance, the text indicates that beta is a constant, instead of a different constant for each variable, which is the case in the cited reference), particularly that of the coupling constant G, and (ii) a discussion of their validity, given that the 2T models used are not identical (in Ref. 19 they have additional e-e and e-ph scattering terms reducing both temperatures) and thus the same effective parameters might not be adequate.

3) The 2T model does not reproduce the long-wavelength features in both BG and NPG, but this is not adequately acknowledged and discussed.

4) The TMM+2TM model shows that the lifetime of the electronic excitation is longer in the NPG, but this is information already coded in the model through the scaling of the coupling constant by a power of the filling factor. At the very least, this choice should be clearly justified.

5) Interpreting the negative values of transmissivity as arising from interband transitions seems reasonable. However, it would be interesting to check if this is the case through other tests. Could this be perhaps tested by measuring photoemission lifetimes?

6) Regarding the study of the spectral response at different pulse fluence, several aspects of the modelling and the critical comparison with the experiment are unclear: (i) the model predicts qualitatively different features than those found

experimentally for the NPG, resembling instead those of the BG (e.g. Fig. 3d), (ii) it predicts a larger redshift than the one seen experimentally which, incidentally, could almost be interpreted as step-wise instead, (iii) it is not clear why Fig. 5b tracks the minima, while in the model (Fig. 4d) shows that the wavelengths crossing zero values are the ones redshifting, while the minima do not, (iv) note S5 does not provide evidence to show that from fluences of 3.5 mJ/cm² the response is nonlinear, as claimed in the text.

7) The presentation of the CL data in Fig. S3 is unclear. The square in Fig. S3a does not seem to correspond with the zoomed data in panel b. Also, the color legend in panel d is unclear, because it only shows labels one color in a map that includes data on a gradient.

8) I might have missed it, but I think that the thickness of the gold layer not noted.

9) The model used for the TMM is not described, except for the choice of permittivities.

10) The time delay scale is not included in Fig. 3ab.

11) The discussion on the CL data (Fig. 2) should be clarified. (i) As far as I understood, the CL data from "holes and gaps" is not shown in the figure, yet the text compares that against the profiles shown in Fig. 2c. (ii) This section of the characterization is not clearly connected with the rest of the discussion; what are the key takeaways of these data and how they connect with the rest of the study?

Reviewer #2

(Remarks to the Author)

This paper reports on ultrafast transient absorption spectroscopy experiments on a nanoporous gold film (NPG) and compares the results to those obtained with the well studied continuous bulk gold (BG) film. The two films display significantly different transient properties, with the NPG having a negative DT/T extending to significantly longer wavelengths and a much longer electron-phonon thermalization time, extending to several picoseconds.

The experimental results are interesting and novel, but I have the impression that the authors are trying to oversell them. First of all it is not clear to me why the authors call NPG a metamaterial: to my understanding, a metamaterial is a material with nanostructured surface volume designed on purpose to have properties which are not found in ordinary materials (see e.g. the seminal paper DOI: 10.1126/science.1210713). Here there is no on purpose design, but just a random nanoporous structure without any control of the properties. The abstract goes even further and defines NPG as a "temporal metamaterial". There is currently a great interest in time-varying materials, which are materials whose physical properties, such as refractive index, change over time, often in a controlled and dynamic way, typically exploiting nonlinear optical effects. This temporal modulation introduces new degrees of freedom for manipulating light waves. Again here I do not see how the NPG can be considered as a time-varying material.

I find the paper to be not very well organized and written, with also several grammar mistakes. For example, figure 1 starts with simulations of electronic temperature using the two-temperature model (TTM) without even showing the experimental data. Also stationary transmission/extinction spectra of the NPG and BG films are not given.

The most striking difference between the BG and the NPG in my opinion is the much longer electron cooling time in the latter sample. However, I miss a clear physical explanation of why electron-phonon thermalization is much longer in the case. Also, the dynamics in Fig. 3c stop at 8 ps, how does the relaxation in NPG proceed at longer delays? The comparison between experimental and simulated DT/T spectra in NPG is not very satisfactory: in the experiment, the positive peak is higher than the negative one, while in theory it is almost zero. This discrepancy is not discussed in the manuscript. The authors have a very good time resolution of sub-15 fs, however they do not focus on the very early stages of electron dynamics, monitoring the electron-electron scattering process resulting in the build-up of a thermalized electron distribution. It would be very interesting to compare this process in BG and NPG.

In summary, the current version of the manuscript does not make a compelling case for publication in Nature Communications. If the authors provide a better organized manuscript with clearer physical insight and toned down claims, then the manuscript could be reconsidered.

Reviewer #3

(Remarks to the Author)

This manuscript focuses on the understanding specific excitation pathways for excited carriers in nanoporous gold as compared to bulk gold. It uses ultrafast cathodoluminescence spectroscopy as well as two-state temperature modeling to understand the properties of these films, additionally a model describing the relationship between the interband and plasmonic excitations is included. They also tune the porosity to create a relationship between fill factor and absorption, and thus electronic temperature and excitation. This work is important and interesting for exciting hot carriers in plasmonic, however I have questions and concerns about its breadth and applicability.

1. It is clear the porosity is important for this property? But why, is this all driven solely by the changes in the mean free path?

2. how can these changes be modeled from either first principles or the dielectric function, to give broader physical insights? Could this be understood by the reductions in the scattering lifetime due to porosity? Presumably that is interrelated with changes in the absorption.

3. How does porosity change the extinction cross-section? Meaning, not just the MFP but also the intrinsic ability to thermalize photons?

4. One concern I have is how to understand and generalize this inverse porous design within the context of what is well known in colloidal and assembled systems. For example, nonporous gold in some sense is an assembly of small gold inclusions or a mesh of gold nanowires, albeit with slightly higher conductivity. This to me is the main gap of the paper as written. How can this result be connect with the broader knowledge of nanoparticle systems? Could experiments and models be designed to integrate these two areas of knowledge? To me, this is absolutely critical for this manuscript.

5. for example, brining the gap to works such as reference 37 and other similar colloidal works focused on excited electronic processes in plasmonics.

Version 1:

Reviewer comments:

Reviewer #1

(Remarks to the Author)

The authors have responded to all my comments in adequate detail and improved the manuscript significantly. I can recommend it for publication, as is, although I recommend the authors to discuss a couple of points in more detail:

1) The methodology used to obtain the results shown in the new Fig. 5, especially as to how do you differentiate between intraband and interband transitions.

2) Expand the critical discussion to justify taking the electron-phonon coupling factor as scaling with the filling factor f . I personally don't find this intuitive or convincing: the volume occupied by the electrons remains the same as that occupied by the phonons. If anything, I would expect the coupling to change with the confinement of the electrons beyond their mean free path, or perhaps with changes in the phonon density of states due to confinement. The dependence may end up being lineal with the filling factor, but the reasoning behind it may change considerably.

Reviewer #2

(Remarks to the Author)

The revised version of the manuscript satisfactorily addresses the comments by myself and by the other reviewers. New experimental and computational data (including atomistic simulations) are added and the explanation of the observed physical phenomena is now much more convincing. Some claims that were not fully justified have now been removed. In the present form, the paper is suitable for publication in Nature Communications.

Reviewer #3

(Remarks to the Author)

i now find the manuscript suitable for publication

Dear Reviewers,

We thank you for the careful evaluation of our manuscript. Guided by your comments, we have revised it. The major changes are:

- **New samples and measurements.** While re-measuring the old samples to answer the Reviewers' questions, we realized that those samples were degraded. Thus, new samples were fabricated and measured to assess the reproducibility of the old measurements and to carefully address the Reviewers' concerns about data interpretation and discussion on the physical origin of the reported results.
- **Revised effective medium theory calculations.** An updated effective medium theory has been implemented by correcting the previous model. We also provide a more detailed description of the relevant assumptions and parameters used in the Methods section. This ensures a thorough explanation of the main mechanism affecting the observed changes in the contributions of electronic transitions to the optical response of nanoporous gold.
- **Atomistic simulations.** We added atomistic simulations to show the respective contributions of interband and intraband transitions to the optical response in both the porous and the bulk samples. These results support the cathodoluminescence data, which were not clearly contextualized in the previous version.
- **Figure order updated.** Following the Reviewers' feedback, in particular that of Reviewer #2, we revised both the figures and the order of discussion points.

We hope that this revised version clarifies that modified interband and intraband transitions contributions observed in nanoporous gold (relative to the bulk gold case) arise from morphology. This conclusion is supported by complementary measurements (ultrafast pump-probe, cathodoluminescence, and ultraviolet photoemission spectroscopies) and by modelling the system with both an effective medium theory and atomistic simulations.

Below are our point-by-point answers to the Reviewers' comments.

REVIEWER COMMENTS

Reviewer #1 (Remarks to the Author):

Tapani et al. studied the transient response of a nanoporous metal gold (NPG) substrate using pump-probe spectroscopy and contrast it with that of a flat gold layer (BG). Their experimental study is explored using a computational model combining transfer matrix

method (TMM) and a two-temperature model (2TM) to account for the absorbance of the system and internal electronic dynamics, respectively. The analysis of the results focuses on the possibility of allowing interband transitions in gold after an initial pulse elevates the effective electronic temperature and consequently creates empty states in the conduction band at energies lower than the Fermi level. The article explores an interesting phenomenon, but I think that it does not explore it in sufficient depth: some key aspects of the characterization do not seem to match the discussion and explanation of the results, the theoretical modelling is not sufficiently well described and fails to explain one of the significant features of the data. Although I would be very interested in seeing a significantly extended version of this study to better understand the phenomenon at hand, I do not think that this manuscript is ready for publication.

Authors' answer. We thank the Reviewer for his/her appreciation of our work and for stating that our article **“explores an interesting phenomenon”**. We agree with the Reviewer that our analysis could have been more detailed, and we hope that our revised manuscript, addresses the Reviewer’s concerns. We also would like to thank the Reviewer for triggering additional investigations, which prove undoubtedly the role of morphology on the observed physical effects.

Reviewer’s comment #1. *The difference between BG and NPG on their initial effective electron temperature after the pulse is argued to arise from a difference in their absorption at the pump's wavelength, 850 nm. However, not only this difference is not shown in the main text or SI, but the spectral data shown in Fig. S2, which is presented as justification for the larger absorption of the NPG, suggests instead that their absorption might not be that different after all. Cathodoluminescence (CL) data, which could perhaps be used as an indirect proxy for the absorbance, does not reach these wavelengths either.*

Authors' answer. We thank the Reviewer for raising this point. As mentioned above, we fabricated new samples and repeated the pump-probe experiments to ensure both full reproducibility and understanding of the underlying physics. NPG is clearly absorbing more than BG at the pump wavelength (see revised Supplementary Figure S6). As discussed later when presenting more details about the two-temperature and the effective medium theory models, this higher absorption, due to the morphology of the sample, is also affecting the initial electronic temperature after pump excitation (see also our answer to Reviewer’s comment #2).

Regarding the CL data, our aim was to show that we have strong localized excitations at the probe wavelengths due to sample morphology. These excitations are mainly due to localized plasmon resonances, which contain modified contributions from both interband and intraband transitions, as proved by our new atomistic simulations.

Action taken. We updated Supplementary Figure S6 to show the extended region up to 840 nm (maximum spectral resolution of our instrument). We added an additional sentence to clarify why we performed CL measurements (see page 7, lines 235-237): “Having shown that nanoporosity modifies the ultrafast optical response of the sample, and that this is connected to a modified interband transition response under femtosecond pump excitation, we investigated whether this is a purely transient effect or whether it also appears in steady-state conditions.”.

Reviewer’s comment #2. *On the 2T model, additional information is needed: (i) explicit details about the values of beta taken from Ref. 19 (for instance, the text indicates that beta is a constant, instead of a different constant for each variable, which is the case in the cited reference), particularly that of the coupling constant G, and (ii) a discussion of their validity, given that the 2T models used are not identically (in Ref. 19 they have additional e-e and e-ph scattering terms reducing both temperatures) and thus the same effective parameters might not be adequate.*

Authors’ answer. We thank the Reviewer for raising this point. Since the last version, we updated the two-temperature model to the extended two-temperature model (e2TM). We also revisited the assumptions for the scaling of the material parameters for NPG and changed to a linear scaling of both the heat capacities and the electron-phonon coupling factor. All details are now in the revised Methods section and in the SI. Nevertheless, we would like to clarify here one important point. Since we treated the NPG as an effective medium, the parameters in equation (2) in the Methods section in the main text, have to be scaled to account for the porosity. The heat capacity of air is negligible compared to that of Au and does not meaningfully contribute to the effective heat capacity of the NPG sample. Since the volumetric heat capacities are used in equation (2), the effective values for the NPG are the Au values scaled with the filling factor f , such that only the Au part of the NPG contributes to the heat capacities. The same reasoning is applied to the electron-phonon coupling factor. Electrons only interact with phonons inside the Au, therefore the electron-phonon coupling factor, which is a volumetric quantity, also scales linearly with f . Here, we neglected surface-mediated energy transfer from electrons to the lattice, which we assume is a secondary effect in our NPG sample that has an average ligament size of approximately 50 nm since the electron-phonon coupling primarily involves short-wavelength phonons (see also new Reference [71]). So, to represent NPG in the e2TM, we scale the parameters as $C_{e,l} \rightarrow fC_{e,l}$ and $G \rightarrow fG$. Mathematically, this is equivalent to scaling the source terms by f^{-1} , which physically means that the energy absorbed from the pump pulse is distributed on less volume of Au, compared to a film. Using the temperatures from the e2TM, we could estimate the permittivity change $\Delta\varepsilon(T_e, T_l)$ induced by the pump.

Action taken. We updated the two-temperature model to the extended two-temperature model and changed to a linear scaling of the effective material parameters for NPG. The Method section and the SI have been updated such that all details are presented, including the discussion on the mathematical and physical reasons why we scale both heat capacity and coupling factors linearly with the filling factor (see page 13, lines 447-467).

Reviewer's comment #3. *The 2T model does not reproduce the long-wavelength features in both BG and NPG, but this is not adequately acknowledged and discussed.*

Authors' answer. As mentioned above, we have fabricated new samples and re-characterized them. We found that the previously observed positive signal originated mainly from strong scattering in both BG and NPG. After re-aligning the beams and repeating measurements at multiple locations, we found that the scattering came from specific spots on the samples, most likely due to dust and/or micrometric surface defects, which are known to give a strong Rayleigh scattering contribution. The new results show that we do not have this effect anymore after new and clean samples were fabricated, in particular for the BG film, which was already characterized by other groups (see for instance Della Valle et al., Phys. Rev. B 86, 155139 (2012)).

Reviewer's comment #4. *The TMM+2TM model shows that the lifetime of the electronic excitation is longer in the NPG, but this is information already coded in the model through the scaling of the coupling constant by a power of the filling factor. At the very least, this choice should be clearly justified.*

Authors' answer. We thank the reviewer for raising this important point. We kindly refer to our answer to the Reviewer's comment #2.

Action taken. We updated the scaling of the parameters in the e2TM for the NPG. We updated the manuscript with more details and justifications for our assumptions (see page 3, lines 111-125, and pages 13, lines 447-467).

Reviewer's comment #5. *Interpreting the negative values of transmissivity as arising from interband transitions seems reasonable. However, it would be interesting to check if this is the case through other tests. Could this be perhaps tested by measuring photoemission lifetimes?*

Authors' answer. Triggered by curiosity and the Reviewer's suggestion, we performed X-ray photoemission spectroscopy (XPS) to probe the valence band of BG and NPG, as discussed in Supplementary Note 6. We found no difference between the linewidth and/or the peak position of the 5d band in the two cases, leading us to the conclusion that the observed effects are not connected to a change in the band structure but to a change in the morphology, which strongly modifies the contributions from intraband and interband transitions to the optical response of the NPG sample. This is also confirmed by the fact that to describe the transient response of the NPG sample we can use an effective medium theory which captures the main physical features. Furthermore, in the revised manuscript, both CL measurements and atomistic modelling independently point to the fact that we also have a modified contribution from intraband and interband transitions due to localized excitations, which are caused by the morphology. Finally, we also tested, using our effective medium theory, if the observed effect has a more trivial origin, such as larger positive transient reflectance, implying that no electronic transitions are promoted by the probe light (see Supplementary Note 5). The transient change of the real and imaginary part of the permittivity upon pump excitation is shown in Supplementary Figure S7. The imaginary part is positive for both BG and NPG, indicating that the observed effects arise from probe light absorption and not from ground state bleaching.

Action taken. We performed new XPS measurements focused more on the valence band spectral region and added a new discussion in SI (see Supplementary Note 6). We updated the discussion on CL measurements (see page 7-8, lines 237-256) and updated old Figure 2, new Figure 4. We performed frequency-dependent fluctuating charges and dipoles (ω FQF μ) atomistic simulations and added a discussion (see page 8-9, lines 268-297) and new Figure 5. We also added a discussion in the SI about the possibility that the observed effect has a more trivial origin, such as larger positive transient reflectance, implying that no electronic transitions are promoted by the probe light (see Supplementary Note 5).

Reviewer's comment #6. *Regarding the study of the spectral response at different pulse fluence, several aspects of the modelling and the critical comparison with the experiment are unclear: (i) the model predicts qualitatively different features than those found experimentally for the NPG, resembling instead those of the BG (e.g. Fig. 3d), (ii) it predicts a larger redshift than the one seen experimentally which, incidentally, could almost be interpreted as step-wise instead, (iii) it is not clear why Fig. 5b tracks the minima, while in the model (Fig. 4d) shows that the wavelengths crossing zero values are the ones redshifting, while the minima do not, (iv) note S5 does not provide evidence to show that from fluences of 3.5 mJ/cm² the response is nonlinear, as claimed in the text.*

Authors' answer. We thank the Reviewer for these insightful comments. We have repeated the measurements on new samples and improved the model. As proved by the new Figure 2 and Figure 3, it is now clear that the model captures the main experimental features. In a new discussion, we directly illustrate and compare the broadening of the negative signal region observed experimentally (see new Figure 3). Our effective medium theory reproduces very well this behaviour.

Action taken. We updated Figure 3 and the discussion (see pages 5-7, lines 160-219). We removed the previous discussion on zero crossing value, local minimum and non-linearity of the signal, as they no longer apply.

Reviewer's comment #7. *The presentation of the CL data in Fig. S3 is unclear. The square in Fig. S3a does not seem to correspond with the zoomed data in panel b. Also, the color legend in panel d is unclear, because it only shows labels one color in a map that includes data on a gradient.*

Authors' answer. We thank the Reviewer for raising this point. We agree that the original figure was not sufficiently clear. We have now moved the CL spectra of the BG to the main text to compare more easily the two cases, that is BG and NPG. New Figure 4a and 4d are the SEM images, while Figures 4b and 4e are the CL intensity maps. The confusion, due to a lack of clarity from our side, raised from the fact that we showed a SEM image and showed the magnified region as a CL intensity map, which was not a magnification of the SEM image. We have now clarified this in the revised version. Also, we have added CL intensity spectra to panel (c) corresponding to the pixels indicated in panel (a). We have also updated the colour bar in panel (b), extending its range to include a wider spectrum of wavelengths. As detailed in the manuscript, the colour corresponds to the wavelength at the maximum of the emission spectra, which is strictly confined between 520 and 550 nm, thus confirming that in BG contributions from interband transitions come mainly around 2.3 eV (L-point in gold band structure).

Action taken. We moved the old Figure 2 and related discussion to a later stage in the discussion of the results (see new Figure 4), and updated its caption (see page 8, line 259-266).

Reviewer's comment #8. *I might have missed it, but I think that the thickness of the gold layer not noted.*

Authors' answer. The thickness of the BG film is 26 nm. Due to fabrication processes (see Kwon, H.; et al. *ACS Appl. Mater. Interfaces* **2023**, *15*, 5620–5627, Ref. 41 in the main

text) the NPG film has a varying thickness around this value (± 4 nm). We have added this information to the Methods section.

Action taken. We added this information in the Methods section (see page 11, lines 362-364): “Both the NPG and BG films have a nominal thickness of 26 nm. Due to its morphology change during fabrication⁴¹, the NPG film has a varying thickness around this value (± 4 nm, that is a maximum of 8 nm from dip to peak).”

Reviewer’s comment #9. *The model used for the TMM is not described, except for the choice of permittivities.*

Authors’ answer. We thank the reviewer for noticing this. The modelling section in the Methods section has been updated accordingly to specify what TMM model we used, that is the Abeles’ matrix method.

Action taken. We updated the Methods section accordingly and added the references we used to build our own model (see page 12, lines 399-418).

Reviewer’s comment #10. *The time delay scale is not included in Fig. 3a,b.*

Authors’ answer. We thank the reviewer for pointing it out. We’ve corrected it.

Action taken. We updated the old Figure 3, which is now part of the new Figure 1.

Reviewer’s comment #11. *The discussion on the CL data (Fig. 2) should be clarified. (i) As far as I understood, the CL data from “holes and gaps” is not shown in the figure, yet the text compares that against the profiles shown in Fig. 2c. (ii) This section of the characterization is not clearly connected with the rest of the discussion; what are the key takeaways of these data and how they connect with the rest of the study?*

Authors’ answer. We thank the Reviewer for this observation. Regarding the first point, we added CL data from a region with a gap (no gold) as shown in new Figures 4d and 4f to show that no emission between 500 and 750 nm is happening from the voids/gaps regions. Regarding the second point: CL measurements has been performed to highlight the presence of localized excitations in NPG compared to BG. These excitations are mainly due to the presence of localized plasmon polaritons which have contributions from both interband and intraband transitions. Nevertheless, CL cannot distinguish

between intraband and interband contributions. This is why we performed atomistic simulations to disentangle the two effects.

Action taken. We updated Figure 2, now new Figure 4. We added a sentence to clarify why we performed CL measurements (see page 7, lines 235-237): “Having shown that nanoporosity modifies the ultrafast optical response of the sample, and that this is connected to a modified interband transition response under femtosecond pump excitation, we investigated whether this is a purely transient effect or whether it also appears in steady-state conditions.”

Reviewer #2 (Remarks to the Author):

This paper reports on ultrafast transient absorption spectroscopy experiments on a nanoporous gold film (NPG) and compares the results to those obtained with the well-studied continuous bulk gold (BG) film. The two films display significantly different transient properties, with the NPG having a negative DT/T extending to significantly longer wavelengths and a much longer electron-phonon thermalization time, extending to several picoseconds. The experimental results are interesting and novel, but I have the impression that the authors are trying to oversell them.

Authors' answer. We thank the Reviewer for his/her positive comments and for stating that the “**experimental results are interesting and novel**”. We also acknowledge the improper use of some terms and claims, which were removed in the revised version (see our answers below).

Reviewer's comment #1. *First of all, it is not clear to me why the authors call NPG a metamaterial: to my understanding, a metamaterial is a material with nanostructured surface volume designed on purpose to have properties which are not found in ordinary materials (see e.g. the seminal paper DOI: 10.1126/science.1210713). Here there is no on purpose design, but just a random nanoporous structure without any control of the properties.*

Authors' answer. We respectfully provide few references to support our claim about NPG as a metamaterial:

- Jalas, D. *et al.* Effective medium model for the spectral properties of nanoporous gold in the visible. *Appl. Phys. Lett.* **105**, 241906 (2014).
- Jalas, D., *et al.* Electrochemical tuning of the optical properties of nanoporous gold. *Sci Rep* 7, 44139 (2017).

- Garoli, D. *et al.* Fractal-Like Plasmonic Metamaterial with a Tailorable Plasma Frequency in the near-Infrared. *ACS Photonics* **5**, 3408–3414 (2018).
- Rout, S., Qi, Z., Biener, M.M. *et al.* Nanoporous gold nanoleaf as tunable metamaterial. *Sci Rep* **11**, 1795 (2021).
- Ebrahimi, F. *et al.* Size-Dependent Photoemission Study by Electrochemical Coarsening of Nanoporous Gold. *J. Phys. Chem. C* **128**, 12960–12968 (2024).

These works by different groups show that NPG structures behave as plasmonic metamaterials whose effective plasma frequency can be tuned by controlling the fractal dimension (ligament/pore morphology), which matches well with the claim “*a metamaterial is a material with nanostructured surface volume designed on purpose to have properties which are not found in ordinary materials*”.

Action taken. We have added the suggested reference by the Reviewer and some of the above-mentioned references in the main text to clarify why NPG can be considered a metamaterial (see page 2, lines 46-49).

Reviewer’s comment #2. *The abstract goes even further and defines NPG as a “temporal metamaterial”. There is currently a great interest in time-varying materials, which are materials whose physical properties, such as refractive index, change over time, often in a controlled and dynamic way, typically exploiting nonlinear optical effects. This temporal modulation introduces new degrees of freedom for manipulating light waves. Again here I do not see how the NPG can be considered as a time-varying material.*

Authors’ answer. We appreciate this comment and find it genuinely helpful to help us to contextualize our work better. We agree that the use of the term “temporal metamaterial” in the abstract and the main text was not appropriate in the context of this work.

Action taken. We removed any reference to “temporal metamaterial” in the manuscript and SI.

Reviewer’s comment #3. *I find the paper to be not very well organized and written, with also several grammar mistakes. For example, figure 1 starts with simulations of electronic temperature using the two-temperature model (TTM) without even showing the experimental data.*

Authors’ answer. We thank the Reviewer for the feedback on manuscript logic organization and writing quality. We have now added experiments in Figure 1 and updated

the other figures and related discussions to make more logical connections between the sections. Now the logic is the following:

(i) We study the transient response of NPG and compare it with the ultrafast dynamics of a continuous gold film using non-degenerate optical pump-probe spectroscopy. We find that the nanoporous sample displays significantly different electron relaxation dynamics directly following pump excitation. This is attributed to a much higher electronic temperature in the nanoporous material, causing a stronger smearing of the Fermi-Dirac distribution.

(ii) The experimental results are well described by an extended two-temperature model, which highlights the role of morphology in enabling the more efficient generation of hot carriers, and the consequent lower-energy photons absorption inducing a modification of the contribution of interband transitions to the optical response of the porous sample.

(iii) Cathodoluminescence spectroscopy reveals the presence of localized plasmon resonances across the visible range which we ascribe to morphology-modified contributions from both intra- and interband transitions.

(iv) Atomistic simulations disentangle intraband and interband contributions to these excitations, pointing out the role of nanoscale porosity in modifying the electronic response of the system compared to the bulk benchmark sample.

Action taken. We updated all the figures and revised the manuscript structure. We also performed a full grammar and wording check on the manuscript.

Reviewer's comment #4. Also, stationary transmission/extinction spectra of the NPG and BG films are not given.

Authors' answer. We thank the Reviewer for raising this point. The spectra from 500 nm to 840 nm is now in the SI.

Action taken. We updated Supplementary Figure S6.

Reviewer's comment #5. *The most striking difference between the BG and the NPG in my opinion is the much longer electron cooling time in the latter sample. However, I miss a clear physical explanation of why electron-phonon thermalization is much longer in the case.*

Authors' answer. We thank the Reviewer for pointing this out. We agree that the explanation needed to be stated more clearly. Fundamentally, this difference stems from a combination of multiple factors: (i) significantly higher absorbed power per Au volume due to the nanoporous morphology and thus higher electron temperature in NPG

compared to BG, and (ii) the heat capacity of Au increases with electronic temperature, and higher heat capacity slows down relaxation dynamics. Thus, the elevated initial electron temperature in NPG compared to BG is responsible for the slower dynamics. See also our answers to Reviewer #1 comments #2 and #4.

Action taken. We updated the manuscript with more details and justifications for our assumptions (see page 3, lines 111-125, and page 13, lines 447-467).

Reviewer's comment #6. *Also, the dynamics in Fig. 3c stop at 8 ps, how does the relaxation in NPG proceed at longer delays?*

Authors' answer. We agree that this is a valuable information for the readers. We have extended the time window to 10 ps in the main text. Additionally, we show extended traces for both NPG and BG up to 20 ps in the SI.

Action taken. We updated old Figure 2 (now part of new Figure 1), and added Supplementary Figure 5.

Reviewer's comment #7. *The comparison between experimental and simulated DT/T spectra in NPG is not very satisfactory: in the experiment, the positive peak is higher than the negative one, while in theory it is almost zero. This discrepancy is not discussed in the manuscript.*

Authors' answer. We kindly refer to our answer to Reviewer #1 comment #3.

Reviewer's comment #8. *The authors have a very good time resolution of sub-15 fs, however they do not focus on the very early stages of electron dynamics, monitoring the electron-electron scattering process resulting in the build-up of a thermalized electron distribution. It would be very interesting to compare this process in BG and NPG.*

Authors' answer. We thank the Reviewer for pointing this out. We compared the two cases, and we do not see a significant difference in the rising phase, as shown in the figure below, where we plot normalized transient transmission $\Delta T/T$ of NPG (yellow curve) and BG (purple curve) films as a function of the time delay (up to 2 ps) between pump and probe pulses at 540 nm. This indicates that the rising phase dynamics are governed primarily by Au's electronic structure rather than geometry, which dominates the thermalization process as captured very well by our e2TM. Additionally, we provide in SI the full spectral and temporal characterizations of the pump and probe pulses.

Action taken. We added Supplementary Figure 3 and 4 (full spectral and temporal characterizations of the pump and probe pulses).

Reviewer’s comment #8. *In summary, the current version of the manuscript does not make a compelling case for publication in Nature Communications. If the authors provide a better organized manuscript with clearer physical insight and toned down claims, then the manuscript could be reconsidered.*

Authors’ answer. We hope the revised version, containing a different structure, revised claims and clearer physical insights, addresses the Reviewer’s concerns.

Reviewer #3 (Remarks to the Author):

This manuscript focuses on the understanding specific excitation pathways for excited carriers in nonporous gold as compared to bulk gold. It uses ultrafast and cathodoluminescence spectroscopy as well as two-state temperature modelling to understand the properties of these films, additionally a model describing the relationship between the interbank and plasmonic excitations is included. They also tune the porosity to create a relationship between fill factor and absorption, and thus electronic temperature and excitation. This work is important and interesting for exciting hot carriers in plasmonics, however I have questions and concerns about its breadth and applicability.

Authors' answer. We thank the Reviewer for his/her positive comments, such as *“this work is important and interesting for exciting hot carriers in plasmonics”*. We also appreciate the Reviewer's concern about the breadth and applicability of our results. In response, we have revised the discussion on these aspects in our manuscript (see also our responses to the Reviewer's concern here below).

Reviewer's comment #1. *It is clear the porosity is important for this property? But why, is this all driven solely by the changes in the mean free path?*

Authors' answer. In our revised manuscript we prove the key role of morphology/porosity in affecting electronic transitions in the NPG system. Nevertheless, we have realized that the explanation for the observed differences needed to be clarified better. We kindly refer to our answers to question #5 by Reviewer #2 and question #2 by Reviewer #1 for a more detailed explanation. We just want to stress here that there are 2 main factors involved for the observed effects in NPG compared to the BG case: (i) a significantly higher absorbed power per Au volume due to the nanoporous morphology and thus higher electron temperature in NPG compared to BG, and (ii) the decrease of the electronic heat capacity and electron-phonon coupling due to the nanoporosity.

Reviewer's comment #2. *How can these changes be modelled from either first principles or the dielectric function, to give broader physical insights? Could this be understood by the reductions in the scattering lifetime due to porosity? Presumably that is interrelated with changes in the absorption.*

Authors' answer. We agree that first-principles simulations would give deeper microscopic insight, but they are not practical at the scale of our NPG networks (thousands of atoms). Instead, we model porosity with a Bruggeman effective-medium description using Rakić's Au permittivity; the metal filling fraction sets the effective dielectric function. Porosity-driven lifetime effects enter automatically through the ϵ_2/TM via filling-fraction scaling. In parallel, we use compact atomistic $\omega\text{FQF}\mu$ calculations on representative ligament-neck motifs to tease apart interband and intraband contributions to the optical response. These atomistic simulations have been shown to provide a very detailed description of plasmonic behavior as detailed as DFT calculations (see Giovannini et al., ACS Photonics 9, 3025–3034 (2022); Giovannini et al., Nanoscale 11, 6004–6015 (2019); Giovannini et al., J. Phys. Chem. Lett. 11, 7595–7602 (2020)).

Reviewer's comment #3. *How does porosity change the extinction cross-section? Meaning, not just the MFP but also the intrinsic ability to thermalize photons?*

Authors' answer. The porosity increases the absorption, specifically below 2.4 eV, by effectively reducing the difference between the refractive index of air and of the sample, which leads to a reduced reflectance. Hence, more light interacts with the sample, and a bigger fraction of the total intensity is absorbed. This effect is captured by our effective medium theory. Since only the metallic part of the NPG absorbs light, the absorbed energy is distributed on less volume compared to that of BG. This leads to significantly higher electron temperatures (see the inset in new Figure 1b).

Reviewer's comment #4. *One concern I have is how to understand and generalize this inverse porous design within the context of what is well known in colloidal and assembled systems. For example, nonporous gold in some sense is an assembly of small gold inclusions or a mesh of gold nanowires, albeit with slightly higher conductivity. This to me is the main gap of the paper as written. How can this result be connected with the broader knowledge of nanoparticle systems? Could experiments and models be designed to integrate these two areas of knowledge? To me, this is absolutely critical for this manuscript.*

Authors' answer. We thank the Reviewer for this important point. The model can theoretically be generalized to any system that can be represented by the Bruggeman effective medium model, meaning mixtures where the filling factor of each material is of a similar magnitude. There is no specific limit to the model, but generally it has been used successfully for two-mixture samples where the filling factor of one material is at least 30%. Furthermore, the Bruggeman model can also be adapted to represent samples where the inclusions are of a specific shape. In the Bruggeman equation (see Methods, equation (1), page 12), the depolarization factors can be altered to represent inclusions of different shapes. Some examples are inclusions in the shape of vertical rods and disk-like inclusions.

Action taken. We added Supplementary Note 7 and Supplementary Figure S9, where we discuss the possibility to represent different systems with Bruggeman model.

Reviewer's comment #5. *For example, bringing the gap to works such as reference 37 and other similar colloidal works focused on excited electronic processes in plasmonics.*

Authors' answer. We thank the Reviewer for this comment. As discussed in the previous answer, the new Supplementary Note 7 has been added to address how the model can be used for different colloidal systems.

Action taken. We added a sentence in the main text (see page 10, lines 328-330): “Interestingly, the applied opto-thermal workflow is controlled by morphology and can be generalized to colloidal or assembled films, suggesting the phenomenon may occur across a broader class of systems (see Supplementary Note 7 and Supplementary Figure S9).”

Dear Reviewers,

We thank you for reviewing our revised manuscript and are pleased to read that the new version has addressed most of the Reviewers' comments. Below we provide our point-by-point answers to the remaining remarks.

We would also like to mention additional minor corrections: we have edited the numbering of the equations (Eqs. 4–12) and corrected a typing error in Eqs. 3 and 4. Furthermore, we added 6 new references that we found during continued discussions on the topic and we added labels in Figure 5c for clarity.

REVIEWER COMMENTS

Reviewer #1 (Remarks to the Author):

The authors have responded to all my comments in adequate detail and improved the manuscript significantly. I can recommend it for publication, as is, although I recommend the authors to discuss a couple of points in more detail:

Authors' answer. We thank the Reviewer for their assessment of the revised manuscript. We have extended the discussion on the additional aspects as follows.

Reviewer's comment #1. *The methodology used to obtain the results shown in the new Fig. 5, especially as to how do you differentiate between intraband and interband transitions.*

Authors' answer. We thank the Reviewer for his/her comment, which allows us to better clarify this point. As reported in more detail in Refs. (Liebsch, Phys. Rev. B **48**, 11317 and Giovannini *et al.*, ACS Photonics 2022, 9, 9, 3025–3034), interband transitions are associated with the polarizability of the d-shell. In the atomistic model $\omega\text{FQ}\mu$, such contributions are taken into account by including induced dipole moments μ , while intraband transitions are described by means of the Drude model of conduction and associated to induced charges q . To quantify the relative weight of inter- and intra-band transitions, we have introduced the averaged quantities $\langle f_q \rangle$, $\langle f_\mu \rangle$ (defined in the Methods section, see Eqs. 11–12, page 15, lines 521–528).

Action taken. We have added the following sentence in the Methods section (see page 14, lines 488–489): "To describe d-metals, the complex dipole moment is introduced to take into account the polarizability of the d-shell (Liebsch, Phys. Rev. B **48**, 11317 and Giovannini *et al.*, ACS Photonics 2022, 9, 9, 3025–3034)". We also clarified the description in the main text (see page 9 lines 285–299).

Reviewer's comment #2. *Expand the critical discussion to justify taking the electron-phonon coupling factor as scaling with the filling factor f . I personally don't find this intuitive or convincing: the volume occupied by the electrons remains the same as that occupied by the phonons. If anything, I would expect the coupling to change with the confinement of the electrons beyond their mean free path, or perhaps with changes in the phonon density of states due to confinement. The dependence may end up being linear with the filling factor, but the reasoning behind it may change considerably.*

Authors' answer. We agree with the Reviewer's opinion that justification of this key assumption in our modelling is important. As described in the Methods section (page 13, lines 450–466), the scaling with the filling factor f in the nanoporous gold film is based on the fact that the e2TM considers volumetric quantities for the heat capacity (units J/m³/K) and the e-ph coupling (W/m³/K). Thus, when we consider an effective material that is a mixture of air and gold, only the gold volume fraction contributes to the heat capacity and e-ph coupling, meaning that the quantities are smaller per unit volume in the effective material than in bulk gold. With this scaling, we assume that the mechanisms from which the electrons and phonons are interacting does not change in the nanoporous sample. This means that we neglect surface-mediated energy transfer from electrons to the lattice, which we assume is a secondary effect in our NPG sample that has an average ligament size of approximately 50 nm. This assumption is founded on the fact that e-ph coupling is connected to electronic transitions involving mainly short wavelength phonons, see also the derivation of the phenomenological e-ph coupling factor (Peierls, R. (1932). Elektronentheorie der Metalle. In: Ergebnisse der exakten naturwissenschaften. Ergebnisse der Exakten Naturwissenschaften, vol 11. Springer, Berlin, Heidelberg. <https://doi.org/10.1007/BFb0111800>). Our results, showing good agreement between model and experiment, support these assumptions. At even smaller ligament sizes, additional effects may indeed no longer be negligible.

Action taken. We have rephrased a sentence in the manuscript (see lines 450–452): “Since we treated the NPG as an effective medium that is a mixture of air and gold, where only the gold volume fraction contributes to the heat capacity and electron-phonon coupling, the parameters in equation (2) had to be scaled to account for the sample morphology.”

Reviewer #2:

The revised version of the manuscript satisfactorily addresses the comments by myself and by the other reviewers. New experimental and computational data (including atomistic simulations) are added and the explanation of the observed physical phenomena is now much more convincing. Some claims that were not fully justified have now been removed. In the present form, the paper is suitable for publication in Nature Communications.

Authors' answer. We thank the Reviewer for his/her final and positive assessment of our work.

Reviewer #3:

I now find the manuscript suitable for publication.

Authors' answer. We thank the Reviewer for his/her final and positive assessment of our work.

Dear Editor, dear Reviewers,

We thank you for reviewing our revised manuscript.

Below we summarize the minor changes we have made to the manuscript in order to fulfill the formal editorial requirements. The relevant sections are also highlighted in yellow in the main manuscript.

- We have corrected affiliations and addresses.
- We have shortened the abstract in accordance with the word limit (150 words).
- We have updated equations, atomic orbital notation, and figures to match the style requirements. In particular we have corrected the schematics in Figure 1 so as to not use any potential third-party content.
- We have corrected formatting of the reference list. We have also updated the data availability statement to include the figshare link and DOI.
- We have included initials of authors in the funding information.

We provide more details regarding all formatting in the attached checklist.